# Effects of Deacetylasperulosidic Acid on Atopic Dermatitis through Modulating Immune Balance and Skin Barrier Function in HaCaT, HMC-1, and EOL-1 Cells

**DOI:** 10.3390/molecules26113298

**Published:** 2021-05-30

**Authors:** Jin Su Oh, Geum Su Seong, Yong Deok Kim, Se Young Choung

**Affiliations:** 1Department of Life and Nanopharmaceutical Sciences, Graduate School, Kyung Hee University, 26 Kyungheedae-ro, Dongdaemun-gu, Seoul 02447, Korea; ok9638@naver.com; 2Korea Food Research Institute, 245 Nongsaengmyeong-ro, Iseo-myeon, Wanju Gun 55365, Jeollabuk-do, Korea; gsseong824@nate.com; 3NST BIO Co., Ltd., Goeumdal-ro, Yangchon-eup, Gimpo-si 10049, Gyeonggi-do, Korea; ydkim@nstbio.co.kr; 4Department of Preventive Pharmacy and Toxicology, College of Pharmacy, Kyung Hee University, 26 Kyungheedae-ro, Dongdaemun-gu, Seoul 02447, Korea

**Keywords:** *Morinda citrifolia*, atopic dermatitis, deacetylasperulosidic acid, HaCaT, HMC-1, EOL-1, immune balance, skin barrier function

## Abstract

The medicinal plant noni (*Morinda citrifolia*) is widely dispersed throughout Southeast Asia, the Caribbean, and Australia. We previously reported that fermented Noni could alleviate atopic dermatitis (AD) by recovering Th1/Th2 immune balance and enhancing skin barrier function induced by 2,4-dinitrochlorobenzene. Noni has a high deacetylasperulosidic acid (DAA) content, whose concentration further increased in fermented noni as an iridoid constituent. This study aimed to determine the anti-AD effects and mechanisms of DAA on HaCaT, HMC-1, and EOL-1 cells. DAA inhibited the gene expression and secretion of AD-related cytokines and chemokines including interleukin (IL)-1β, IL-4, IL-6, IL-8, IL-25, IL-33, thymic stromal lymphopoietin, tumor necrosis factor-alpha, monocyte chemoattractant protein-1, thymus and activation-regulated chemokine, macrophage-derived chemokine, and regulated upon activation, normal T cell expressed and secreted, in all cells, and inhibited histamine release in HMC-1 cells. DAA controlled mitogen-activated protein kinase phosphorylation levels and the translocation of nuclear factor-kappa light chain enhancer of activated B cells into the nucleus by inhibiting IκBα decomposition in all the cells. Furthermore, DAA increased the expression of proteins involved in skin barrier functions such as filaggrin and involucrin in HaCaT cells. These results confirmed that DAA could relieve AD by controlling immune balance and recovering skin barrier function.

## 1. Introduction

Atopic dermatitis (AD) is a chronic skin disease that affects many people worldwide [1]. AD generally occurs due to a combination of environmental, genetic, and immunological factors [2,3]. It is accompanied by eczema, erythema, xeroderma, lichenification, and persistent itching [4]. A characteristic feature of AD is an increase in the number of T helper 2 (Th2) cells [5,6]. The increase in the level of cytokines released from Th2 cells in AD causes a Th1/Th2 immune imbalance and leads to a lower expression of skin barrier proteins [7]. It has been reported that damage to the skin barrier function is closely related to AD [8,9,10,11,12,13] as it can worsen AD through inflammatory reactions related to AD [4]. Therefore, the restoration of Th1/Th2 immune balance and skin barrier function is the objective of AD treatment. Corticosteroids are the most commonly used drugs for the treatment of AD [14]. However, corticosteroids can cause various side effects, such as xeroderma, acne, and depression [15]. Therefore, the development of anti-AD drugs with fewer side effects is necessary.

Keratinocytes maintain physical and biochemical conditions, as well as supply moisture to the skin. Keratinocytes can cause immune responses in AD through the mitogen-activated protein kinase (MAPK) and nuclear factor kappa-light-chain-enhancer of activated B cells (NF-kB) pathways [16,17]. They generate and release cytokines such as interleukin (IL)-25, IL-33, and thymic stromal lymphopoietin (TSLP) [18], which can induce activation of the MAPK and NF-kB pathways in mast cells and eosinophils [19].

NF-κB increases the expression of AD-related cytokines and chemokines such as IL-1β, IL-6, IL-8, tumor necrosis factor-α (TNF-α), and monocyte chemoattractant protein-1 (MCP-1) [16,20]. Therefore, the control of MAPK and NF-κB signaling is vital for AD treatment. Keratinocytes in AD also increase the expression and release of chemokines such as thymus and activation-regulated chemokine (TARC) and macrophage-derived chemokine (MDC) [16,19,21]. TARC and MDC are connected to C-C chemokine receptor type 4 (CCR4) on Th2 cells and are involved in the differentiation of Th2 cells [22,23]. Increases in TARC and MDC level were found in the sera of patients with AD [24,25,26]. Therefore, TARC and MDC are important indices for the diagnosis and treatment of AD. Regulated on activation, normal T cell expressed and secreted (RANTES) expressed and secreted by keratinocytes can induce eosinophil activation. It is also involved in chronic AD by inducing the migration and invasion of Th2 cells to the inflammatory site [27]. In addition, keratinocytes can generate proteins such as filaggrin (FLG) and involucrin (IVL), which are involved in skin barrier function [28]. IL-4, IL-13, and IL-31, cytokines released from the Th2 cells, can lead to severe AD and cause damage to skin barrier proteins [29]. The loss of skin barrier function might allow increased exposure to various antigens and further worsens AD by causing clinical symptoms and excessive inflammatory responses related to AD. Histamine secreted from mast cells can cause itching, which increases inflammatory responses related to AD in keratinocytes and worsens AD-mediated damage of skin barrier function [6]. AD-related cytokines and chemokines such as IL-1β, IL-4, IL-6, IL-8, TSLP, TNF-α, and MCP-1 are expressed and secreted by mast cells [16,30,31], which can stimulate eosinophils and keratinocytes and activate Th2 cells [32].

Eosinophils mainly play defensive roles against parasites and increases in their levels are observed in the peripheral blood of patients with AD [33]. IL-5 released from mast cells and Th2 cells is involved in the growth and survival of eosinophils [34]. By moving toward inflamed areas, activated eosinophils can express and secrete cytokines and chemokines related to AD, such as IL-6, IL-8, TNF-α, and MCP-1, which might worsen AD symptoms [35,36]. AD-related cytokines and chemokines released from eosinophils can stimulate mast cells and keratinocytes, leading to the activation of Th2 cells. Keratinocytes, mast cells, and eosinophils play essential roles in both acute and chronic AD. Therefore, these three cells are essential in studying the anti-atopic effects and mechanisms [37].

Noni (*Morinda citrifolia*), which is widely dispersed throughout Southeast Asia, the Caribbean, and Australia, has been used as a medicinal plant [38]. It contains active ingredients such as flavonoids and iridoids and has been used as a folk remedy for neurological disorders, cancer, diabetes, and inflammation for a long time [39,40,41]. A previous study reported that noni could improve many clinical symptoms related to AD in an AD animal model induced by 2,4-dinitrochlorobenzene. Additionally, it has been demonstrated that noni can restore the Th1/Th2 immune balance, thus having beneficial effects on AD. The purpose of this study was to investigate the anti-atopic effect and mechanism of deacetylasperulosidic acid (DAA), a functional component present in high amounts in fermented noni. Three cells known to play key roles in AD symptoms were used in this study: HaCaT human keratinocyte cells, HMC-1 human mast cells, and EOL-1 human eosinophilic leukemia cells.

## 2. Results

### 2.1. HPLC-PDA Chromatogram of Noni and F. noni

HPLC-PDA analysis at 235 nm was used to determine the components of noni and f. noni, and the established standard chromatogram is shown in Figure 1. DAA and asperulosidic acid (AA), the major constituents of *Morinda citrifolia*, were detected at high concentrations in noni and f. noni. The retention time (Rt; DAA, 14.95 ± 0.08 min; AA, 23.27 ± 0.07 min) matched with the standard. Content analysis indicated that the Noni extract powder contained 10.37 ± 0.94 mg/g of DAA and 0.98 ± 0.08 mg/g of AA. F. noni extract powder contained 13.11 ± 0.25 mg/g of DAA and 1.40 ± 0.08 mg/g of AA (Table 1).

### 2.2. DAA and AA Inhibited the Secretion of AD-Related Inflammatory Cytokines and Chemokines Increased by TNF-α and IFN-γ in HaCaT Cells

The amounts of DAA and AA were increased in fermented noni (Figure 1 and Table 1). The anti-atopic efficacy of DAA and AA was evaluated using HaCaT cells to determine the active ingredients. First, the non-cytotoxic concentrations of DAA and AA to HaCaT cells were determined. Concentrations of DAA and AA up to 0.2 µM did not affect HaCaT cell viability (Figure 2A).

The amounts of TSLP and IL-33 secreted by HaCaT cells were investigated to identify the active ingredients. The secretion of TSLP (Figure 2B) and IL-33 (Figure 2C) was significantly increased in the control group. DAA and AA similarly inhibited the secretion of TSLP and IL-33, and the inhibitory effects of the 0.2 µM DAA and AA were similar to those of 0.1 µM dexamethasone (DEX).

### 2.3. DAA and AA Inhibited the Secretion of AD-Related Inflammatory Cytokines and Chemokines Increased by PMACI in HMC-1 Cells

The anti-atopic efficacy of DAA and AA was investigated in HMC-1 cells to identify the active ingredients. First, the non-cytotoxic concentrations of DAA and AA to HMC-1 cells were determined. Concentrations of DAA and AA up to 0.2 µM did not affect HMC-1 cell viability (Figure 3A). The amount of histamine and IL-4 released from HMC-1 cells was then determined.

The results showed that the amount of histamine (Figure 3B) and IL-4 (Figure 3C) was significantly increased in the control group. DAA and AA similarly inhibited the secretion of histamine and IL-4, and cells treated with the 0.2 µM DAA and AA showed inhibitory effects similar to those treated with 0.1 µM DEX.

### 2.4. DAA and AA Inhibited the Secretion of AD-Related Inflammatory Cytokines and Chemokines Increased by House Dust Mite (HDM) in EOL-1 Cells

The anti-atopic efficacy of DAA and AA was evaluated in EOL-1 cells to identify the active ingredients. First, the non-cytotoxic concentrations of DAA and AA to EOL-1 cells were determined. DAA and AA concentrations up to 60 nM did not affect the EOL-1 cell viability (Figure 4A).

The amounts of MCP-1 and IL-5 released by EOL-1 cells were then evaluated. The results showed that the levels of MCP-1 (Figure 4B) and IL-5 (Figure 4C) were significantly increased in the control. However, these levels were inhibited by DAA and AA in a concentration-dependent manner.

DAA and AA similarly inhibited the secretion of MCP-1 and IL-5, and cells treated with 60 nM DAA and AA showed inhibitory effects similar to those treated by 2 nM DEX. These results show that DAA and AA similarly inhibited AD-related cytokines and chemokines in all the cells used in this study. In addition, DAA and AA similarly inhibited histamine release in HMC-1 cells. Since DAA was higher in content than AA after fermentation, it was selected as the active ingredient for further analysis, and anti-atopic activity and mechanisms were investigated.

### 2.5. DAA Inhibited the Secretion of AD-Related Cytokines and Chemokines by TNF-α and IFN-γ Treatment in HaCaT Cells

The inhibitory effect of DAA on the secretion of AD-related cytokines and chemokines was investigated in the supernatant of HaCaT cells treated with TNF-α and IFN-γ (Figure 5A–E). The levels of IL-1β, IL-6, IL-8, TNF-α, and MCP-1 secreted were significantly higher in the control group than in the normal group. Cytokine and chemokine secretion was inhibited in a concentration-dependent manner. The inhibitory effect of 0.2 µM DAA was similar to that of the 0.1 µM DEX.

### 2.6. DAA Inhibited the Gene Expression Levels of AD-Related Cytokines and Chemokines Increased by TNF-α and IFN-γ in HaCaT Cells

RT-qPCR analysis was performed to investigate the expression levels of AD-related cytokines and chemokines in HaCaT cells treated with TNF-α and IFN-γ (Figure 6A–E). The gene expression of cytokines and chemokines IL-1β, IL-6, IL-8, TNF-α, and MCP-1 was significantly higher in the control group than in the normal group. DAA treatment suppressed this expression in a concentration-dependent manner. In addition, 0.2 μM DAA showed an inhibitory effect similar to 0.1 μM DEX.

### 2.7. DAA Inhibited AD-Related Cytokines and Chemokines Known to Activate Th2 Cells Secreted by HaCaT Cells Treated with TNF-α and IFN-γ

The effect of DAA on the secretion of cytokines and chemokines that activate Th2 cells was investigated in the supernatant of HaCaT cells treated with TNF-α and IFN-γ (Figure 7A–D). The levels of cytokine IL-25 and chemokines TARC, RANTES, and MDC secreted were significantly higher in the control group than in the normal group. Cytokine and chemokine secretion were inhibited in a concentration-dependent manner. The inhibitory effect of 0.2 µM DAA was similar to that of the 0.1 µM DEX.

### 2.8. DAA Inhibited the Gene Expression Levels of AD-Related Cytokines and Chemokines Known to Increase Th2 Activity in HaCaT Cells Treated with TNF-α and IFN-γ

RT-qPCR analysis was performed to investigate the inhibitory effect of DAA on the gene expression of AD-related cytokines and chemokines that activate Th2 cells in HaCaT cells treated with TNF-α and IFN-γ (Figure 8A–E).

The gene expression of cytokines IL-25, IL-33, and TSLP, and chemokines TARC, MDC, and RANTES were significantly increased in the control group. DAA treatment inhibited AD-related cytokine and chemokine levels in a concentration-dependent manner. In addition, 0.2 μM DAA showed an inhibitory effect similar to that of the 0.1 μM DEX.

### 2.9. DAA Inhibited MAPK Phosphorylation and NF-κB Activation Increased by TNF-α and IFN-γ in HaCaT Cells

The phosphorylation levels of MAPK in HaCaT cells treated with TNF-α and IFN-γ were significantly increased in the control group compared to the normal group (Figure 9A(A-a–A-c)). The phosphorylation of MAPK was inhibited in a concentration-dependent manner following DAA treatment. Additionally, a higher inhibitory effect was seen in cells treated with 0.2 µM DAA than in those treated with 0.1 µM DEX.

Furthermore, IκBα decomposition and NF-κB translocation into the nucleus were significantly increased in the control group compared to the normal group, which were inhibited in a concentration-dependent manner after DAA treatment. DAA treatment inhibited the decomposition of IκBα and translocation of NF-κB into the nucleus (Figure 9B(a,b)). The inhibitory effects on IκBα degradation and NF-κB translocation were similar in 0.2 µM DAA and 0.1 µM DEX groups.

### 2.10. DAA Increased the Expression of Skin Barrier Proteins Reduced by TNF-α and IFN-γ in HaCaT Cells

The expression of FLG and IVL was significantly lower in the control than in the normal HaCaT cells. The expression of FLG and IVL proteins increased in a concentration-dependent manner following DAA treatment (Figure 10A–C). DAA increased the expression of skin barrier protein in a concentration-dependent manner, but 0.1 μM DEX showed similar results as the control.

### 2.11. DAA Inhibited the Secreted Levels of AD-Related Cytokines and Chemokines Increased by PMACI in HMC-1 Cells

The effect of DAA on the secretion of AD-related cytokines and chemokines in the supernatant of HMC-1 cells treated with PMACI was investigated (Figure 11A–F). The secretion of IL-1β, IL-6, IL-8, TNF-α, TSLP, and MCP-1 was significantly higher in the control group than in the normal group. The secretion of cytokines and chemokines was inhibited in a concentration-dependent manner following DAA treatment. In addition, 0.2 μM DAA showed an inhibitory effect similar to 0.1 μM DEX.

### 2.12. DAA Inhibited the Gene Expression Levels of AD-Related Cytokines and Chemokines Increased by PMACI in HMC-1 Cells

RT-qPCR was performed to determine the gene expression levels of AD-related cytokines and chemokines in HMC-1 cells treated with PMACI (Figure 12A–F). The gene expression levels of AD-related cytokines and chemokines such as IL-1β, IL-6, IL-8, TNF-α, TSLP, and MCP-1 were significantly higher in the control group than in the normal group. The gene expression levels of AD-related cytokines and chemokines were inhibited in a concentration-dependent manner by DAA. In addition, 0.2 μM DAA showed an inhibitory effect similar to 0.1 μM DEX.

### 2.13. DAA Inhibited MAPK Phosphorylation and NF-κB Activation Increased by PMACI in HMC-1 Cells

The phosphorylation of MAPK in HMC-1 cells treated with PMACI was significantly increased in the control group compared to the normal group (Figure 13A(A-a–A-c)). The phosphorylation of MAPK was inhibited in a concentration-dependent manner following DAA treatment. DAA treatment at a concentration of 0.2 µM showed a stronger inhibitory effect than 0.1 µM DEX.

In addition, IκBα decomposition and NF-κB translocation into the nucleus were significantly increased in the control group. DAA treatment inhibited the decomposition of IκBα and translocation of NF-κB into the nucleus (Figure 13B(B-a,B-b)). Cells treated with 0.2 µM DAA showed a higher efficacy than 0.1 µM DEX in inhibiting IκBα decomposition and NF-κB translocation into the nucleus.

### 2.14. DAA Inhibited the Secretion of AD-Related Cytokines and Chemokines Increased by HDM in EOL-1 Cells

The effect of DAA on the secretion of AD-related cytokines and chemokines in the supernatant of EOL-1 cells treated with HDM was investigated (Figure 14A–C). The secretion of IL-6, IL-8, and TNF-α was significantly increased in the control group compared to the normal group. DAA inhibited cytokine and chemokine secretion in a concentration-dependent manner. DAA at a concentration of 60 nM showed an inhibitory effect similar to 2 nM DEX.

### 2.15. DAA Inhibited the Gene Expression of AD-Related Cytokines and Chemokines Increased by HDM in EOL-1 Cells

RT-qPCR analysis was performed to confirm the gene expression of AD-related cytokines and chemokines in EOL-1 cells treated with HDM (Figure 15A–D). The gene expression of AD-related cytokines and chemokines IL-6, IL-8, TNF-α, and MCP-1 was significantly increased in the control group compared to the normal group. DAA treatment inhibited the expression of AD-related cytokines and chemokines in a concentration-dependent manner. DAA at a concentration of 60 nM showed an inhibitory effect similar to 2 nM DEX.

### 2.16. DAA Inhibited MAPK Phosphorylation and NF-κB Activation Increased by HDM in EOL-1 Cells

The phosphorylation of MAPK in EOL-1 cells treated with HDM was significantly higher in the control group than in the normal group. The phosphorylation of MAPK was inhibited in a concentration-dependent manner following DAA treatment. Treatment with 60 nM DAA was more inhibitory than treatment with 2 nM DEX (Figure 16A(A-a–A-c)).

In addition, IκBα decomposition and the translocation of NF-κB into the nucleus of EOL-1 cells were significantly increased in the control group compared to the normal group, which was inhibited by DAA treatment (Figure 16B(B-a,B-b)). The inhibition of IkBα decomposition and NF-κB translocation into the nucleus was greater in cells treated with 60 nM DAA than in cells treated with 2 nM DEX.

## 3. Discussion

AD causes an imbalance in the Th1/Th2 immune cells due to an increase in the level of cytokines secreted from Th2 cells, followed by the biased differentiation of Th2 cells. These changes cause complicated pathophysiological effects and defects in skin barrier function [4,42,43]. A previous study demonstrated that fermented Noni had anti-atopic effects in an animal model with lesions similar to AD [4]. In the present study, HPLC-PDA analysis confirmed that the DAA and AA levels in fermented noni were increased by approximately 1.3-and 1.4 times, respectively, compared to those in unfermented noni (Figure 1 and Table 1). DAA and AA are iridoid ingredients present at high concentrations in Noni [44]. It has been reported that AA could inhibit the gene expression levels of IL-6 and TNF-α in RAW264.7 cells treated with lipopolysaccharide [45]. However, there are no studies on the inhibition and mechanism of AD-related cytokines and chemokines secreted from HaCaT, HMC-1, and EOL-1 cells treated with DAA and AA. HaCaT cells maintain a normal keratinocyte shape and are used to investigate the role of keratinocytes in skin diseases related to AD [16]. HMC-1 cells are derived from leukemia patients and are suitable for studies related to mast cells because they have the characteristics of mast cells [46,47]. EOL-1 cells differentiate into eosinophils upon stimulation and are therefore used as an appropriate model for investigating eosinophil function [48]. Therefore, in this study, since the DAA and AA content were relatively high upon fermentation, it was selected as a candidate of functional component, and the anti-atopic efficacy and mechanism were evaluated using AD-related cell lines such as HaCaT, HMC-1, and EOL-1. Keratinocytes can react to foreign antigens and produce TSLP and IL-33, which activate mast cells and eosinophils [18]. TNF-α and IFN-γ can increase the expression of cytokines and chemokines through MAPK and NF-κB activities in HaCaT cells [19]. DAA and AA inhibited TSLP and IL-33, which were secreted by HaCaT cells treated with TNF-α and IFN-γ, in a concentration-dependent manner (Figure 2C). Mast cells in AD are involved in allergic inflammatory reactions [49]. The increased number of mast cells in patients with AD suggests their involvement in AD progression [50,51]. When mast cells are activated by antigens, physiologically active substances such as histamine are released and the expression levels of cytokines and chemokines are increased [14,16,30,31,52]. IgE production in B cells is increased by IL-4 from mast cells. Then IgE degranulates mast cells and released histamine causing itching [32]. PMACI activates mast cells to increase the expression of AD-related cytokines and chemokines as well as the secretion of physiologically active substances [53]. DAA and AA inhibited histamine and IL-4, which were released from HMC-1 cells treated with PMACI in a concentration-dependent manner (Figure 3C). HDM activates eosinophils and increases the expression of cytokines and chemokines related to AD [35,54]. MCP-1, which is mainly secreted by eosinophils, was reported to stimulate not only the expression of IL-4 in T cells but also the activation of Th2 cells in mice [55]. Eosinophils can also increase their lifespan through autocrine signaling of IL-5 [56]. MCP-1 and IL-5 secreted by EOL-1 cells treated with HDM were inhibited by DAA and AA treatment in a concentration-dependent manner (Figure 4C). DAA and AA showed similar inhibitory effects in all the cells used in this study (Figure 2, Figure 3 and Figure 4). DAA and AA levels were similarly increased by fermentation. However, the content of DAA was higher than that of AA. Therefore, DAA was selected to study the anti-atopic effects and mechanisms in HaCaT, HMC-1, and EOL-1 cells.

AD-related cytokines and chemokines, which are expressed and secreted in AD-induced mast cells, can cause acute and chronic AD. The indirect activation of Th2 cells is induced by the activation of eosinophils and mast cells [57]. In the present study, DAA inhibited the protein secretion and gene expression of IL-1β, IL-6, IL-8, TNF-α, and MCP-1 in HaCaT cells in a concentration-dependent manner (Figure 5 and Figure 6). The cytokines IL-25, IL-33, and TSLP, which are released from AD-induced keratinocytes, are increased and either directly or indirectly increase the activation of Th2 cells through the activation of mast cells, eosinophils, dendritic cells, and Langerhans cells [18,58,59]. Chemokines including TARC, MDC, and RANTES are involved in the activation and invasion of Th2 cells [19,24,25,27]. Thus, an imbalance in Th1/Th2 immune cells occurs, worsening AD. In TNF-α- and IFN-γ-treated HaCaT cells, DAA inhibited the protein secretion and gene expression of cytokines, including IL-25, IL-33, and TSLP, and chemokines, including TARC, MDC, and RANTES, in a concentration-dependent manner (Figure 7 and Figure 8). HaCaT cells treated with 0.2 µM DAA showed an efficacy similar to those treated with 0.1 µM DEX, which is used for the clinical treatment of AD. The results in HaCaT cells suggest that the inhibition of cytokines and chemokines by DAA is involved in the inactivation of mast cells and the recovery of immune balance.

The inhibitory effect of DAA on the expression of skin barrier proteins in HaCaT cells was evaluated. Many studies have reported defects in skin barrier function related to AD [4,42]. IL-4, IL-13, and IL-31 released from Th2 cells decrease the expression of FLG and IVL [29]. The expression of FLG is usually expressed in keratinocytes and is decreased by TSLP [4,29]. FLG and IVL play important roles in the skin barrier function. FLG is essential for the structure and function of the stratum corneum and plays a role in host defense against pathogens and allergens. It is also involved in homeostasis, both externally and internally [60,61]. IVL is responsible for the external wall of the epidermis and consists of a skin barrier cross-coupled with various skin barrier proteins, including FLG [62]. This suggests that the recovery of the skin barrier can play an important role in relieving AD. A recent study reported that TSLP receptors exist in the C fibers of primary sensory afferent neurons, and the binding of TSLP to TSLP receptors causes itching [63]. The expressions of FLG and IVL in HaCaT cells treated with TNF-α and IFN-γ were increased by DAA treatment in a concentration-dependent manner. This result was identical to the concentration-dependent decrease in cytokine and chemokine levels, including TSLP, following DAA treatment of HaCaT cells (Figure 2 (A-a), Figure 3, Figure 4, Figure 5 and Figure 6). It also corresponds to the decrease in cytokine levels originating from Th2 cells and the increase in FLG and IVL level in an animal model in a previous study [4]. However, the expression of skin barrier proteins in DEX-treated cells was not significantly different from that in cells treated with TNF-α and IFN-γ. (Figure 10). DEX used in clinical treatment of AD is associated with side effects that cause skin barrier malfunctions [28,64]. Therefore, unlike corticosteroids, the results of this study suggest that DAA increased the restoration of skin barrier protein functions and could help relieve AD.

Degranulated mast cells produce AD-related cytokines and chemokines, which activate Th2 cells and stimulate keratinocytes and eosinophils. Additionally, mast cells release histamine, which causes itching [32]. DAA inhibited histamine secretion in a concentration-dependent manner in HMC-1 cells treated with PMACI. DAA at a concentration of 0.2 µM showed efficacy similar to 0.1 µM DEX (Figure 3B (B-a)). Inhibition of histamine release indicates that DAA may reduce itching in patients with AD. In addition, DAA inhibited the protein secretion and gene expression of IL-1β, IL-6, IL-8, TNF-α, TSLP, and MCP-1 in HMC-1 cells treated with PMACI. At a concentration of 0.2 µM, DAA showed efficacy similar to that of DEX at a concentration of 0.1 µM (Figure 11 and Figure 12). The inhibition of cytokines and chemokines in mast cells showed that DAA could control the activation of Th2 cells, keratinocytes, and eosinophils. Thus, it could relieve AD, followed by the recovery of Th1/Th2 immune balance and skin barrier function.

Eosinophils worsen AD through an allergic reaction, and eosinophilia occurs in patients with AD [65,66]. IL-6, IL-8, TNF-α, and MCP-1 produced in EOL-1 cells stimulated by HDM trigger indirect activation of Th2 cells and the progression of immune responses associated with chronic AD [35]. In the present study, DAA inhibited the protein secretion and gene expression of IL-6, IL-8, TNF-α, and MCP-1 in EOL-1 cells treated with HDM. The efficacy of DAA at a concentration of 60 nM was similar to that of 2 nM DEX (Figure 14 and Figure 15). Since IL-5 in AD sustains the differentiation and survival of eosinophils, an increase in Il-5 expression was observed in AD patients [33,34]. The consistent differentiation and presence of eosinophils due to IL-5 might lead to chronic AD and atopic asthma [67,68]. This study investigated the amount of IL-5 secreted by EOL-1 cells stimulated with HDM. DAA inhibited the secretion of IL-5 in EOL-1 cells stimulated with HDM (Figure 4B (B-a)). Therefore, the inhibition of AD-related cytokines and chemokines, including IL-5 in eosinophils demonstrated that DAA could inhibit the activation of Th2 cells and relieve AD, followed by Th1/Th2 immune balance.

The MAPK and NF-κB pathways of HaCaT, HMC-1, and EOL-1 cells are important for regulating the inflammatory response in AD [16,19,30,69,70]. The MAPK pathway regulates various cell functions, such as cell activation and degranulation. It is also involved in inflammatory reactions by activating NF-κB [16]. NF-κB is closely associated with the gene expression levels of cytokines and chemokines related to AD [20]. IkBα and NF-κB proteins are bound to and present in the cytoplasm. The IκBα protein is degraded by MAPK phosphorylation, and NF-κB is translocated into the nucleus to initiate transcription of the target gene [16,71]. Therefore, the expression of cytokines and chemokines is associated with the activation of MAPKs and NF-κB. In the present study, DAA inhibited the phosphorylation of MAPK in HaCaT, HMC-1, and EOL-1 cells in a concentration-dependent manner. It controlled the decomposition of IκBα and the translocation of NF-κB into the nucleus of HaCaT, HMC-1, and EOL-1 cells. When treated with the highest concentration of DAA in each cell, the phosphorylation of MAPK was inhibited more than that by DEX treatment. Furthermore, the decomposition of IkBα and inhibition of NF-κB translocation showed a similar efficacy to DEX treatment when treated with the highest concentration of DAA in each cell (Figure 9, Figure 13 and Figure 16). DAA controlled MAPK and NF-κB in HaCaT, HMC-1, and EOL-1 cells to inhibit the expression and secretion of AD-related cytokines and chemokines. The inhibition of these cytokines and chemokines means that the Th1/Th2 immune balance can be restored by controlling the activation of Th2 cells. Additionally, the inhibition of MAPK phosphorylation in keratinocytes was reported to increase FLG and IVL level [72]. In the present study, DAA concentration dependently controlled MAPK phosphorylation in HaCaT cells treated with TNF-α and IFN-γ (Figure 9A). This result was identical to the increased expression of FLG and IVL in HaCaT cells (Figure 10). The increase in the level of FLG and IVL, which are skin barrier proteins, is consistent with the decreased expression and secretion of cytokines and chemokines in association with skin barrier damage of keratinocytes, mast cells, and eosinophils. Therefore, the results of this study suggest that DAA is a potential anti-AD drug candidate that can restore Th1/Th2 immune balance and enhance skin barrier function.

## 4. Materials and Methods

### 4.1. Reagents

Deacetylasperulosidic acid (DAA) and asperulosidic acid (AA) were provided by Phytolab (Vestenbergsgreuth, Germany). Isocove’s modified Dulbecco’s medium (IMDM), Roswell Park Memorial Institute-1640 (RPMI-1640), and Dulbecco’s modified Eagle medium (DMEM) were purchased from Gibco BRL (Grand Island, NY, USA). Fetal bovine serum (FBS) was purchased from Gemini Bio (West Sacramento, CA, USA). Dimethyl sulfoxide (DMSO), dexamethasone (DEX), house dust mite (HDM), TNF-α, interferon gamma (IFN-γ), PMA, A23187, and 3-(4,5-dimethylthiazol-2-yl)-2,5-diphenyltetrazolium bromide (MTT) were purchased from Sigma-Aldrich, Inc. (St. Louis, MO, USA). A Cell Counting Kit-8 (CCK-8) was purchased from Dojindo, Inc. (Kumamoto, Japan). IL-1β, IL-4, IL-5, IL-6, IL-8, IL-25, IL-33, TSLP, TNF-α, and MCP-1 were obtained from R&D Systems, Inc. (Minneapolis, MN, USA). TARC, MDC, RANTES, and histamine were purchased from Elabscience, Inc. (Houston, TX, USA). The RNA-spin Total RNA extraction kit was purchased from INtRon Biotechnology (Seoul, Korea). Oligo dT primers were purchased from Bionics Inc. (Seoul, Korea). The cDNA synthesis kit, SYBR green, and rox dyes were purchased from Takara Korea Biomedical, Inc. (Shiga, Japan). The nuclear extraction kit was purchased from Abcam Inc. (Cambridge, UK). Sodium dodecyl sulfate (SDS)-sulfate-polyacrylamide gel was purchased from Bio-Rad (Hercules, CA, USA). The Pierce™ BCA Protein Assay Kit was purchased from Thermo Fisher Scientific, Inc. (Rockford, IL, USA). Anti-phospho-ERK, anti-JNK, -P38, anti-β-actin, anti-lamin B1, -ERK, -JNK, -P38, -IκBα, -NF-κB, -filaggrin, and -involucrin antibodies were obtained from Santa Cruz Biotechnology, Inc. (Dallas, TX, USA). Horseradish peroxidase-conjugated secondary antibodies were obtained from GeneTex, Inc. (Irvine, CA, USA).

### 4.2. Preparation of Fermented Noni Extract (F. noni)

*Morinda citrifolia* (noni) fruit was collected from the NST Bio Noni Farm Co. Ltd in French Polynesia (Indonesia islands), and F.NONI was produced in the NST bio (Gimpo, Korea). Briefly, for the preparation of fermented noni, fresh noni harvested in Indonesia (Java Island) was used. After washing to remove bacteria and foreign substances from fresh noni, they were frozen at −27 °C. The frozen noni stored was thawed at room temperature for two days. For the preparation of fermented noni, 2% probiotic NST 1805 (*Lactobacillus plantarum*) was incubated at 30 °C for 24 to 72 h, heated at 90 °C for 30 min, evaporated and concentrated, dried, and powdered (A2gen). The fruit was thoroughly washed in lukewarm water to remove fungi and to retard the growth of microbiological organisms (yeast, bacteria, etc.) that were sensitive to heat. The noni fruit was then cut into pieces to fit into a 2000 mL container, and 500 mL of water was added. The chunked noni fruit was left to ferment for 24 to 72 h at 30 °C. At the end of the fermentation period, 400 mL of noni fermentation stock was transferred to a 1000 mL container. The ferment stock (400 mL) was filtered using Whatman filter paper and vacuum filtration to eliminate debris and fruit particles from the stock solution. Thirty milliliters of the filtered ferment was used for qualitative phytochemical analysis, and 10 mL was used for PDA spectroscopic analysis and optical density determination. The remaining 360 mL was used for the experiments.

### 4.3. Quantification of DAA and AA by HPLC-PDA Analysis

One gram of the f. noni powder was diluted with 50 mL of H_2_O-MeOH (1:1) and mixed thoroughly, and the solution was collected into a 50 mL volumetric flask for HPLC-PDA analysis. The extracts were combined, filtered, and dried in a rotary evaporator under vacuum at 50 °C. The dried extracts were re-dissolved in MeOH for HPLC analysis. An HPLC system (Shimadzu Corporation, Kyoto, Japan) equipped with an LC-20AD series pumping system and an SPD-M20A photodiode array detector (PDA) was used to analyze Noni, F. noni, and the standard iridoid (DAA and AA). Separation was carried out on a symmetry column (250 × 4.6 mm, 5 μm). The binary mobile phase consisted of water containing 0.1% formic acid (solvent A) and acetonitrile (solvent B). The flow rate was kept constant at 1.0 mL/min for a total run time of 40 min. The mobile phase was run with a gradient program: 0–5 min, 0% B; 5–40 min, 35% B. The sample injection volume was 5 µL. The column temperature was maintained at 25 °C. Peaks of interest were monitored at 190–380 nm using a PDA detector, and the spectra were compared with that of the standard.

### 4.4. Cell Culture

The human keratinocyte cell line (HaCaT), human mast cell line (HMC-1), and human eosinophilic leukemia cell line (EOL-1) were obtained from the Korea Cell Line Bank, Inc. (Seoul, Korea). HaCaT cells were grown in DMEM containing 10% FBS and penicillin (100 units/mL)/streptomycin (100 μg/mL) in an incubator at 37 °C and 5% CO_2_. HMC-1 cells were grown in IMDM medium containing 10% FBS, penicillin (100 units/mL), and streptomycin (100 μg/mL) in an incubator at 37 °C and 5% CO_2_. The EOL-1 cells were grown in RPMI-1640 medium containing 10% FBS, penicillin (100 units/mL), and streptomycin (100 μg/mL) in an incubator at 37 °C and 5% CO_2_.

### 4.5. Cell Viability

The viability of HaCaT and HMC-1 cells was evaluated using the MTT assay. Briefly, HaCaT cells were seeded into 24-well plates at a density of 1 × 10^6^ cells/mL and incubated for 24 h. The HaCaT cells were treated with DAA and AA at concentrations of 0.01, 0.05, 0.1, and 0.2 μM for 48 h. HMC-1 cells were seeded into 24-well plates at a density of 1 × 10^6^ cells/mL and incubated for 24 h. The HMC-1 cells were treated with DAA and AA at concentrations of 0.01, 0.05, 0.1, and 0.2 μM for 24 h, and the cell supernatant was aspirated. After adding 1 mL of MTT (5 mg/mL) solution to each well, the plate was incubated at 37 °C for 5 h. The MTT solution was aspirated and the crystallized formazan was dissolved in DMSO. Optical density (OD) was measured at 565 nm using an enzyme-linked immunosorbent assay (ELISA) reader (Versa Max, Molecular Devices, Sunnyvale, CA, USA). EOL-1 cells were inoculated into 96-well plates at a density of 5 × 10^4^ cells/mL and incubated for 24 h. EOL-1 cells were treated with DAA and AA at concentrations of 3, 15, 30, and 60 nM for 48 h. Next, 10 μL of CCK-8 solution was added to each well, the plate was incubated at 37 °C for 1 h, and the OD was measured at 450 nm using an ELISA reader.

### 4.6. Cytokine and Chemokine Analyses

HaCaT cells were seeded into 24-well plates at a density of 1 × 10^6^ cells/mL and incubated for 24 h. Then, they were pre-treated with DAA at concentrations of 0.01, 0.05, 0.1, and 0.2 μM for 1 h, and treated with TNF-α and IFN-γ, each at a concentration of 10 ng/mL for 24 h. HMC-1 cells were seeded into 24-well plates at a density of 1 × 10^6^ cells/mL and incubated for 24 h. Then, they were pre-treated with DAA at concentrations of 0.01, 0.05, 0.1, and 0.2 μM for 1 h, followed by treatment with PMACI (PMA (40 nM) + A23187 (1 μM)) for 5 h. EOL-1 cells were seeded into 96-well plates at a density of 5 × 10^4^ cells/mL and incubated for 24 h. They were then pre-treated with DAA at 3, 15, 30, and 60 nM for 1 h, followed by treatment with HDM (10 ng/mL) for 24 h. DEX was used as the positive control. HaCaT and HMC-1 cells were treated with DEX at a concentration of 0.1 μM. EOL-1 cells were used at a concentration of 2 nM. The levels of IL-1β, IL-4, IL-5, IL-6, IL-8, IL-25, IL-33, TSLP, TNF-α, MCP-1, TARC, MDC, and RANTES in the HaCaT, HMC-1, and EOL-1 cell supernatants were determined using ELISA kits according to the manufacturer’s instructions (R&D Systems, Inc., Minneapolis, MN, USA,; Elabscience, Inc. (Houston, TX, USA). 

### 4.7. Histamine Analysis

HMC-1 cells were inoculated into 24-well plates at a density of 1 × 10^6^ cells/mL and incubated for 24 h. The cells were pre-treated with DAA for 1 h at concentrations of 0.01, 0.05, 0.1, and 0.2 μM. Positive control cells were treated with 0.1 μM DAA. The cells were then treated with PMA (40 nM) + A23187 (1 μM) for 8 h and the cell supernatants were collected. Histamine levels in the HMC-1 cell supernatants were determined using ELISA kit according to the manufacturer’s instructions.

### 4.8. RNA Extraction and Quantitative Real Time-Polymerase Chain Reaction (RT-qPCR)

HaCaT cells were inoculated into 24-well plates at a density of 1 × 10^6^ cells/mL and incubated for 24 h. The cells were pretreated with DAA (0.01, 0.05, 0.1, and 0.2 μM) for 1 h, and then treated with TNF-α and IFN-γ at a concentration of 10 ng/mL for 24 h. Positive control cells were treated with 0.1 μM DEX. HMC-1 cells were inoculated into 6-well plates at a cell density of 1 × 10^6^ cells/mL and incubated for 24 h. They were then pretreated with DAA at 0.01, 0.05, 0.1, and 0.2 μM for 1 h, followed by treatment with PMACI (PMA (40 nM) + A23187 (1 μM)) for 5 h. As a positive control, the cells were treated with 0.1 μM DEX. EOL-1 cells were seeded into 6-well plates at a density of 5 × 10^4^ cells/mL and incubated for 24 h. They were then pretreated with DAA at 3, 15, 30, and 60 nM for 1 h, followed by treatment with HDM (10 ng/mL) for 24 h. DEX was used as a positive control at a concentration of 2 nM.

After removing the supernatant, HaCaT cells were washed thrice with cold phosphate-buffered saline (PBS). After washing, the RNA was extracted using a total RNA extraction kit. Briefly, HMC-1 and EOL-1 cells were centrifuged at 1000× *g* for 5 min at 4 °C. After removing the supernatant, cold PBS was used to resuspend the cell pellet, followed by centrifugation at 1000× *g* for 5 min at 4 °C. The supernatant was removed, and the cell pellet was used for RNA extraction using a total RNA extraction kit. The extracted RNA was then used for cDNA synthesis using a cDNA synthesis kit and oligo dT primers. The synthesized cDNA was then used for RT-qPCR in an ABI StepOnePlus™ real-time PCR system (Applied Biosystems, Waltham, MA, USA) with SYBR Premix EX Taq. The sequences of the primers used for RT-qPCR are listed in Table 2. The mRNA expression levels were normalized to that of GAPDH using the 2^−ΔΔC t^ method, based on the cycle threshold (Ct) value.

### 4.9. Protein Extraction and Western Blotting

HaCaT cells were seeded into 24-well plates at a density of 1 × 10^6^ cells/mL and incubated for 24 h. They were pretreated with DAA at 0.01, 0.05, 0.1, and 0.2 μM for 1 h, and treated with TNF-α and IFN-γ each, at a concentration of 10 ng/mL for 24 h. DEX was used as a positive control at a concentration of 0.1 μM. HMC-1 cells were seeded into 6-well plates at a density of 1 × 10^6^ cells/mL and incubated for 24 h. Then, they were pretreated with DAA at 0.01, 0.05, 0.1, and 0.2 μM for 1 h, followed by treatment with PMACI (PMA (40 nM) + A23187 (1 μM)) for 5 h. DEX was used as a positive control at a concentration of 0.1 μM. EOL-1 cells were seeded into 6-well plates at a cell density of 5 × 10^4^ cells/mL and incubated for 24 h. They were then pretreated with DAA at 3, 15, 30, and 60 nM for 1 h, followed by treatment with HDM (10 ng/mL) for 24 h. DEX was used as a positive control at a concentration of 2 nM.

After removing the culture supernatant, HaCaT cells were washed thrice with cold PBS. Nuclei were separated using a nuclear extraction kit. Proteins were extracted using a cell lysis buffer. After sonication, the cell lysates were centrifuged at 3000× *g* for 10 min at 4 °C, and the supernatant was collected.

HMC-1 and EOL-1 cells were centrifuged at 1000× *g* for 5 min at 4 °C. After removing the culture supernatant, the cells were washed with cold PBS. After centrifugation at 1000× *g* for 5 min at 4 °C, the supernatant was removed, and the nuclei were separated using a nuclear extraction kit. Proteins were extracted using a cell lysis buffer. After sonication, the cell lysates were centrifuged at 3000× *g* for 10 min at 4 °C, and the supernatant was collected. The cellular protein concentration in the supernatant was quantified using a BCA Protein Assay Kit (Thermo Fisher Scientific, Rockford, IL, USA). Protein samples were then subjected to 12% or 7% sodium dodecyl sulfate-polyacrylamide gel electrophoresis (SDS-PAGE). Proteins separated were then transferred to polyvinylidene fluoride (PVDF) membranes and blocked with 5% skim milk at ~20–25 °C for 1 h. After washing with Tris-buffered saline containing 0.5% Tween-20 (TBST), the membranes were incubated with primary antibody (1:1000 dilution) at 4 °C. The next day, the membranes were incubated with horseradish peroxidase-conjugated secondary antibody (1:5000 dilution) at room temperature for 2 h. The protein bands were visualized using the ChemiDoc™ XRS system (Bio-Rad, Richmond, CA, USA). The expression levels of the target proteins were analyzed using Image Lab statistical software (Bio-Rad) and normalized to those of β-actin and lamin B1.

### 4.10. Statistical Analysis

Data are shown as the mean ± standard deviation (SD). All statistical analyses were performed using one-way analysis of variance and Tukey’s honestly significant difference test. Statistically significant differences were evaluated using SPSS software (SPSS Inc., Chicago, IL, USA). # *p* < 0.05, ## *p* < 0.01, and ### *p* < 0.001 compared to the normal; * *p* < 0.05, ** *p* < 0.01, and *** *p* < 0.001 compared to the control.

## 5. Conclusions

DAA demonstrated anti-atopic activity by inhibiting gene expression and secretion of AD-related cytokines and chemokines in HaCaT, HMC-1, and EOL-1 cells. DAA regulated the expression of AD-related cytokines and chemokines by controlling MAPK and NF-kB pathways in HaCaT, HMC-1, and EOL-1 cells. In addition, DAA increased the expression of skin barrier proteins FLG and IVL in HaCaT cells. Taken together, this study suggests that DAA, the main functional ingredient in fermented noni, may be a therapeutic agent that can alleviate AD by regulating immune balance and restoring skin barrier function.

## Figures and Tables

**Figure 1 molecules-26-03298-f001:**
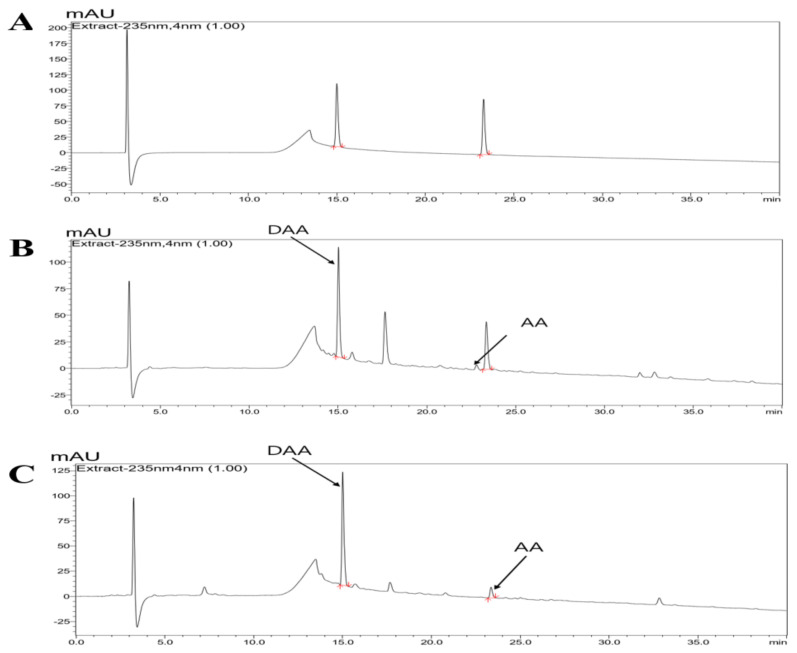
Representative HPLC-PDA chromatogram of noni and f. noni. (**A**) Standard; (**B**) noni extract powder; and (**C**) F. noni extract powder. The arrows represent DAA and AA (standard DDA Rt: 14.95 ± 0.08 min, noni and f. noni each 15.01 ± 0.04 and 15.01 ± 0.03 min; standard AA Rt: 23.27 ± 0.07 min, noni and f. noni each 23.34 ± 0.06 and 23.34 ± 0.04 min).

**Figure 2 molecules-26-03298-f002:**
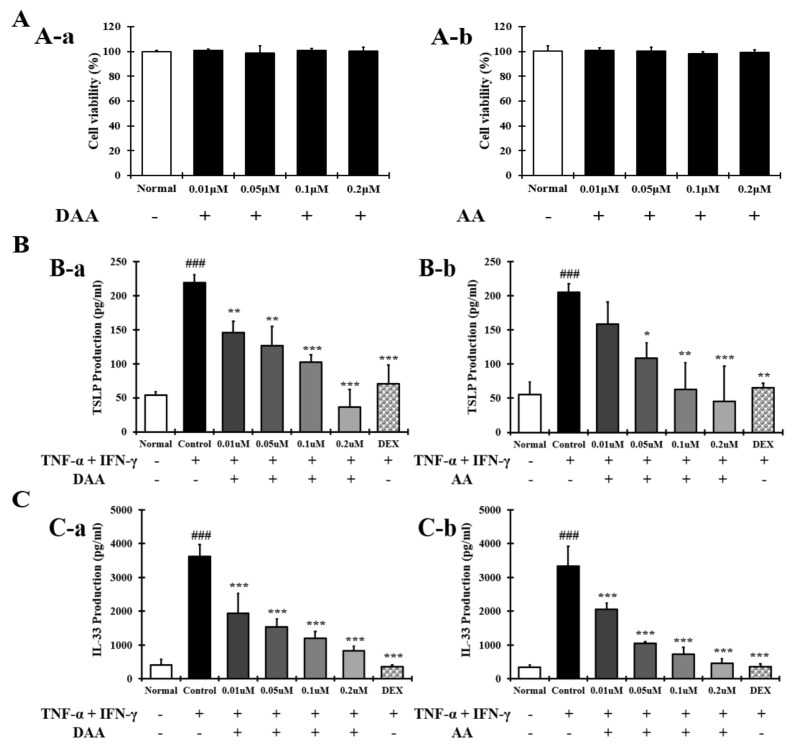
Effect of DAA and AA on HaCaT cell viability and the levels of AD-related cytokines and chemokines in HaCaT cells. (**A**) The viability of HaCaT cells treated with DAA and AA. (**B**) DAA and AA inhibited the secretion of TSLP by HaCaT cells treated with both TNF-α and IFN-γ. (**C**) DAA and AA inhibited the secretion of IL-33 by HaCaT cells treated with both TNF-α and IFN-γ. The production of cytokines and chemokines in the supernatant of HaCaT cells was analyzed by ELISA. The results are expressed as the mean ± SD (n = 4). ### *p* < 0.001 vs. normal; * *p* < 0.05, ** *p* < 0.01 and *** *p* < 0.001 vs. control; DAA was used at 0.01 µM, 0.05 µM, 0.1 µM, and 0.2 µM. DEX, the positive control, was used at 0.1 µM.

**Figure 3 molecules-26-03298-f003:**
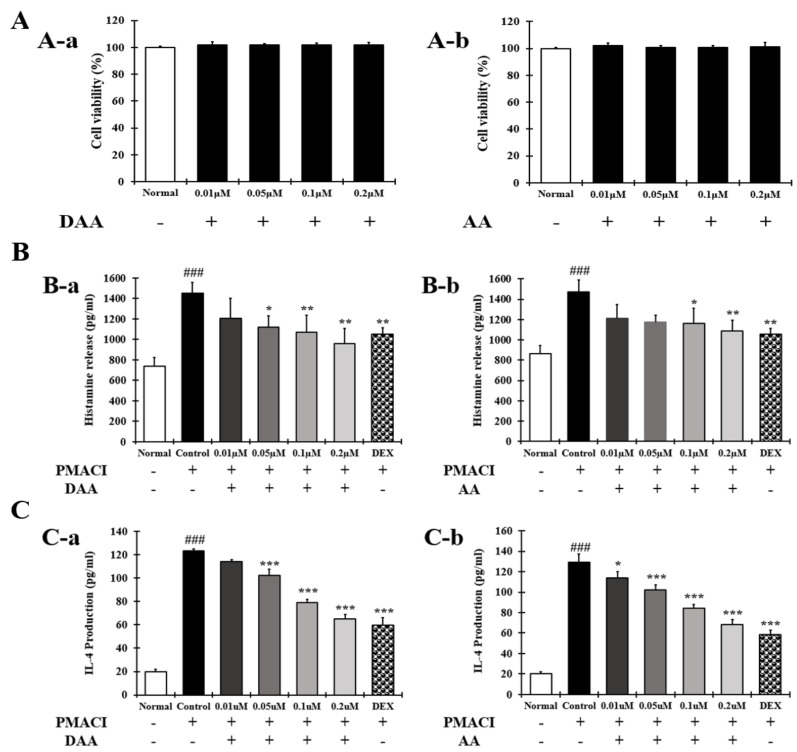
Effect of DAA and AA on HMC-1 cell viability and the levels of AD-related histamine and IL-4 in HMC-1 cells. (**A**) Viability of HMC-1 cells treated with DAA and AA. (**B**) Inhibitory effects of DAA and AA on the levels of histamine secreted by PMACI-treated HMC-1 cells. (**C**) Inhibitory effects of DAA and AA on the levels of IL-4 secreted by PMACI-treated HMC-1 cells. The levels of cytokines and chemokines in the supernatant of HMC-1 cells were determined by ELISA. The results are expressed as the mean ± SD (n = 4). ### *p* < 0.001 vs. normal; * *p* < 0.05, ** *p* < 0.01, and *** *p* < 0.001 vs. control; DAA was used at 0.01 µM, 0.05 µM, 0.1 µM, and 0.2 µM. DEX, the positive control, was used at 0.1 µM.

**Figure 4 molecules-26-03298-f004:**
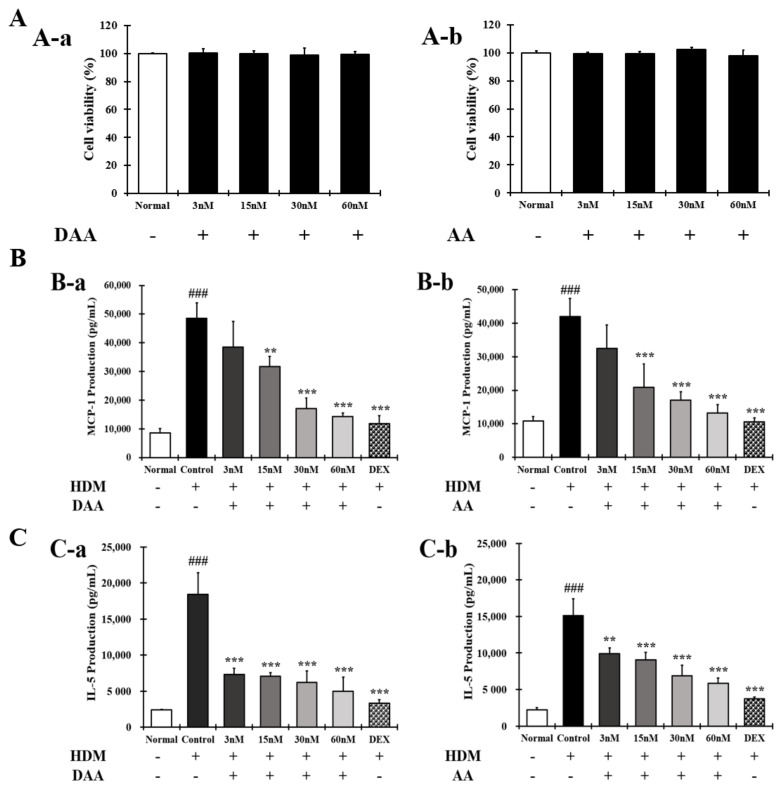
Effect of DAA and AA on the viability of EOL-1 cells and AD-related cytokines and chemokines secreted from EOL-1 cells. (**A**) Viability of EOL-1 cells treated with DAA and AA. (**B**) Inhibitory effects of DAA and AA on the levels of MCP-1 secreted from HDM-treated EOL-1 cells. (**C**) Inhibitory effects of DAA and AA on the levels of IL-5 secreted from HDM-treated EOL-1 cells. The levels of cytokines and chemokines in the supernatant of EOL-1 cells were determined by ELISA. The results are expressed as the mean ± SD (n = 4). ### *p* < 0.001 vs. normal. ** *p* < 0.01, *** *p* < 0.001 vs. control; DAA was used at 3 nM, 15 nM, 30 nM, and 60 nM. DEX, the positive control, was used at 2 nM.

**Figure 5 molecules-26-03298-f005:**
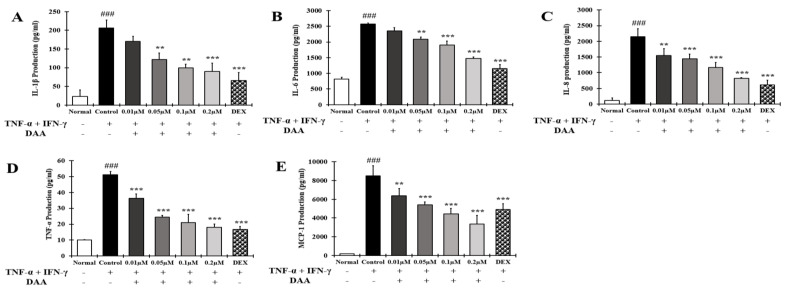
Effect of DAA on AD-related cytokines and chemokines secreted by HaCaT cells. The levels of IL-1β (**A**), IL-6 (**B**), IL-8 (**C**), TNF-α (**D**), and MCP-1 (**E**) in the supernatant of HaCaT cells were evaluated by ELISA. The results are expressed as the mean ± SD (n = 4). ### *p* < 0.001 vs. normal; ** *p* < 0.01 and *** *p* < 0.001 vs. control; DAA was used at 0.01 µM, 0.05 µM, 0.1 µM, and 0.2 µM. DEX, the positive control, was used at 0.1 µM.

**Figure 6 molecules-26-03298-f006:**
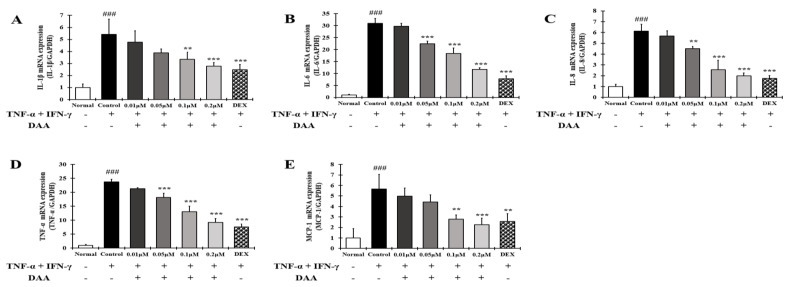
Effect of DAA on the gene expression levels of AD-related cytokines and chemokines in HaCaT cells. The gene expression levels of IL-1β (**A**), IL-6 (**B**), IL-8 (**C**), TNF-α (**D**), and MCP-1 (**E**) are shown. Total RNA was isolated from HaCaT cells and analyzed by RT-qPCR. The expression levels were then normalized to those of GAPDH. The results are expressed as the mean ± SD (n = 4). ### *p* < 0.001 vs. normal; ** *p* < 0.01 and *** *p* < 0.001 vs. control; DAA was used at 0.01 µM, 0.05 µM, 0.1 µM, and 0.2 µM. DEX, the positive control, was used at 0.1 µM.

**Figure 7 molecules-26-03298-f007:**
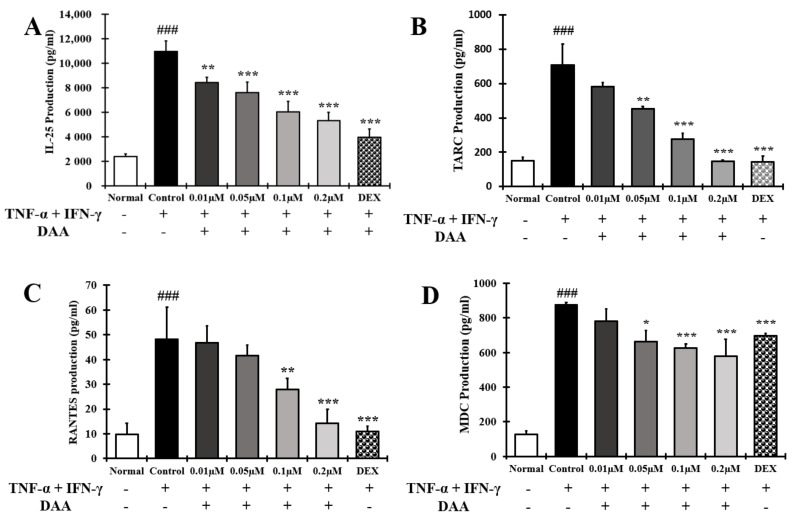
Effect of DAA on the levels of AD-related cytokines and chemokines known to activate Th2 cells secreted by HaCaT cells. The level of cytokine IL-25 (**A**) and chemokines TARC (**B**), RANTES (**C**), and MDC (**D**) in the supernatant of HaCaT cells was evaluated by ELISA. The results are expressed as the mean ± SD (n = 4). ### *p* < 0.001 vs. normal. * *p* < 0.05; ** *p* < 0.01, and *** *p* < 0.001 vs. control; DAA was used at 0.01 µM, 0.05 µM, 0.1 µM, and 0.2 µM. DEX, the positive control, was used at 0.1 µM.

**Figure 8 molecules-26-03298-f008:**
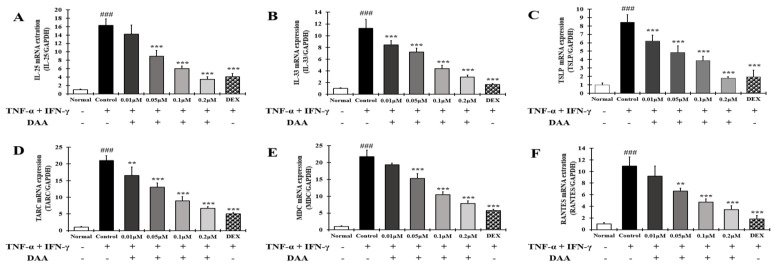
Effect of DAA on the gene expression levels of AD-related cytokines and chemokines involved in Th2 cell activity in HaCaT cells. The gene expression levels of IL-25 (**A**), IL-33 (**B**), TSLP (**C**), TARC (**D**), MDC (**E**), and RANTES (**F**) are shown. Total RNA was isolated from HaCaT cells and analyzed by RT-qPCR. The expression levels were normalized to that of GAPDH. The results are expressed as the mean ± SD (n = 4). ### *p* < 0.001 vs. normal; ** *p* < 0.01 and *** *p* < 0.001 vs. control; DAA was used at 0.01 µM, 0.05 µM, 0.1 µM, and 0.2 µM. DEX, the positive control, was used at 0.1 µM.

**Figure 9 molecules-26-03298-f009:**
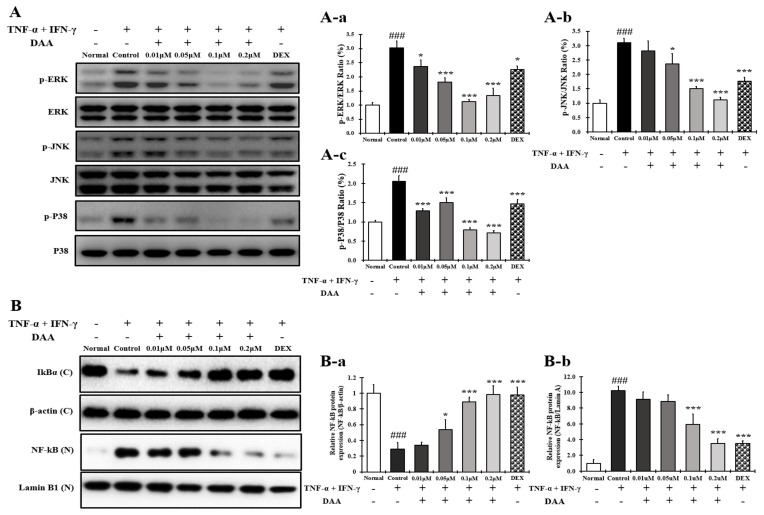
Effect of DAA on MAPK phosphorylation and NF-κB activation in HaCaT cells. (**A**) DAA inhibited the phosphorylation of MAPK (ERK, JNK, and P38) in HaCaT cells (**A-a**–**A-c**). (**B**) DAA also inhibited the degradation of IkBα and the translocation of NF-kB to the nucleus (**B-a**,**B-b**). IκBα was normalized to β-actin and NF-kB was normalized to Lamin B1. The results are expressed as the mean ± SD (n = 4). ### *p* < 0.001 vs. normal; * *p* < 0.05, *** *p* < 0.001 vs. control. DAA was used at 0.01 µM, 0.05 µM, 0.1 µM, and 0.2 µM. DEX, the positive control, was used at 0.1 µM.

**Figure 10 molecules-26-03298-f010:**
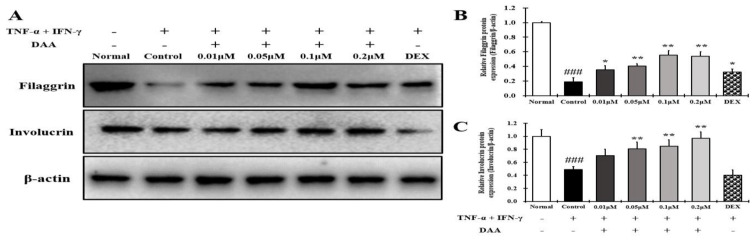
Effect of DAA on the protein expression levels of filaggrin (FLG) and involucrin (IVL) in HaCaT cells. (**A**) DAA increased the expression of filaggrin and involucrin. (**B**) Quantified values of filaggrin. (**C**) Quantified values of involucrin. The levels of filaggrin and involucrin were normalized to that of β-actin. The results are expressed as the mean ± SD (n = 4). ### *p* < 0.001 vs. normal; * *p* < 0.05 and ** *p* < 0.01 vs. control. DAA was used at 0.01 µM, 0.05 µM, 0.1 µM, and 0.2 µM. DEX, the positive control, was used at 0.1 µM.

**Figure 11 molecules-26-03298-f011:**
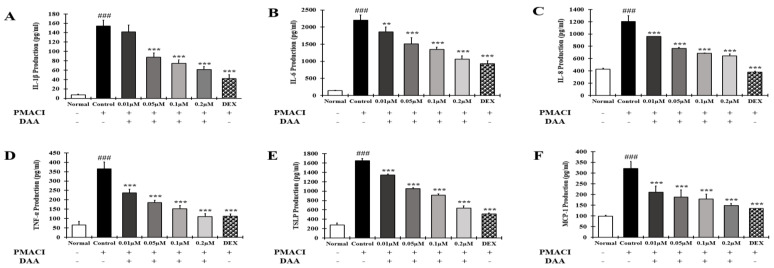
Effect of DAA on AD-related cytokines and chemokines secreted by HMC-1cells. The level of IL-1β (**A**), IL-6 (**B**), IL-8 (**C**), TNF-α (**D**), TSLP (**E**), and MCP-1 (**F**) in the supernatant of HMC-1 cells was determined by ELISA. The results are expressed as the mean ± SD (n = 4). ### *p* < 0.001 vs. normal; ** *p* < 0.01 and *** *p* < 0.001 vs. control; DAA was used at 0.01 µM, 0.05 µM, 0.1 µM, and 0.2 µM. DEX, the positive control, was used at 0.1 µM.

**Figure 12 molecules-26-03298-f012:**
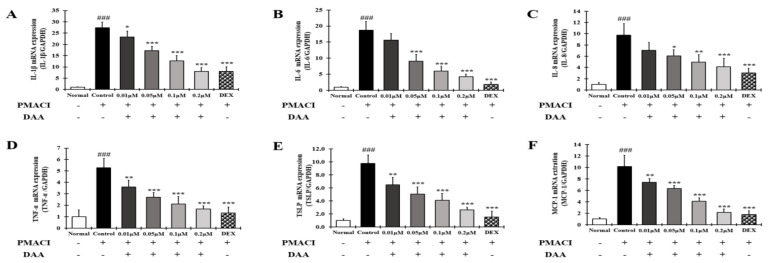
Effect of DAA on the gene expression levels of AD-related cytokines and chemokines in HMC-1 cells. The gene expression levels of IL-1β (**A**), IL-6 (**B**), IL-8 (**C**), TNF-α (**D**), TSLP (**E**), and MCP-1 (**F**) are shown. Total RNA was isolated from HMC-1 cells and analyzed by RT-qPCR. The expression levels were normalized to that of GAPDH. The results are expressed as the mean ± SD (n = 4). ### *p* < 0.001 vs. normal; * *p* < 0.05, ** *p* < 0.01 and *** *p* < 0.001 vs. control; DAA was used at 0.01 µM, 0.05 µM, 0.1 µM, and 0.2 µM. DEX, the positive control, was used at 0.1 µM.

**Figure 13 molecules-26-03298-f013:**
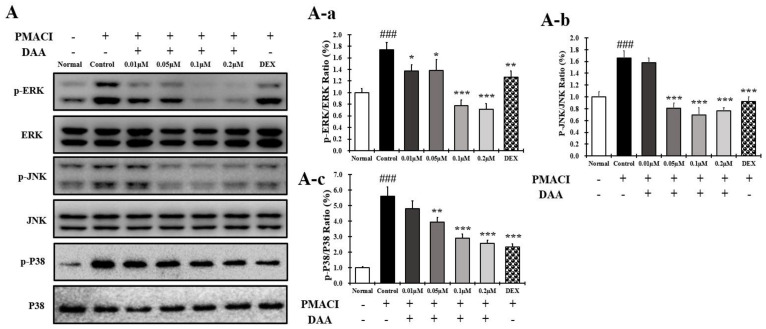
Effect of DAA on MAPK phosphorylation and NF-kB activation in HMC-1 cells. (**A**) DAA inhibited the phosphorylation of MAPK (ERK, JNK, and P38) in HMC-1 cells (**A-a**–**A-c**). (**B**) DAA also inhibited the degradation of IkBα and the translocation of NF-kB into the nucleus (**B-a**,**B-b**). IκBα was normalized to β-actin and NF-kB was normalized to Lamin B1. The results are expressed as the mean ± SD (n = 4). ### *p* < 0.001 vs. normal; * *p* < 0.05, ** *p* < 0.01, and *** *p* < 0.001 vs. control; DAA was used at 0.01 µM, 0.05 µM, 0.1 µM, and 0.2 µM. DEX, the positive control, was used at 0.1 µM.

**Figure 14 molecules-26-03298-f014:**
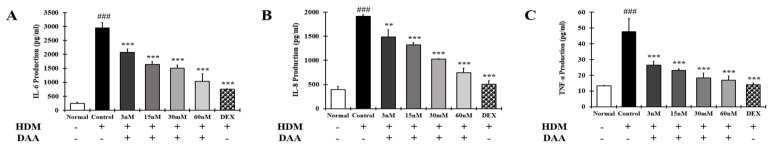
Effect of DAA on AD-related cytokine and chemokine secretion in EOL-1cells. The levels of IL-6 (**A**), IL-8 (**B**), and TNF-α (**C**) in the supernatant of EOL-1 cells were analyzed by ELISA. The results are expressed as the mean ± SD (n = 4). ### *p* < 0.001 vs. normal. ** *p* < 0.01 and *** *p* < 0.001 vs. control. DAA was used at 3 nM, 15 nM, 30 nM, and 60 nM. DEX, the positive control, was used at 2 nM.

**Figure 15 molecules-26-03298-f015:**
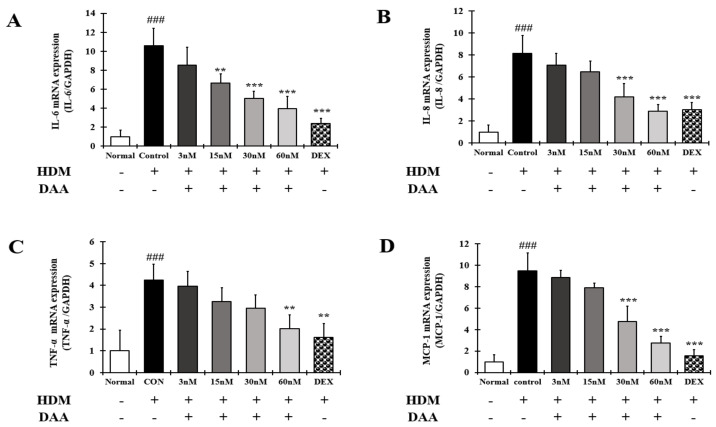
Effect of DAA on the gene expression levels of AD-related cytokines and chemokines in EOL-1 cells. The gene expression levels of IL-6 (**A**), IL-8 (**B**), TNF-α (**C**), and MCP-1 (**D**) are shown. Total RNA was isolated from EOL-1 cells and analyzed by RT-qPCR. The expression levels were normalized to that of GAPDH. The results are expressed as the mean ± SD (n = 4). ### *p* < 0.001 vs. normal; ** *p* < 0.01 and *** *p* < 0.001 vs. control. DAA was used at 3 nM, 15 nM, 30 nM, and 60 nM. DEX, the positive control, was used at 2 nM.

**Figure 16 molecules-26-03298-f016:**
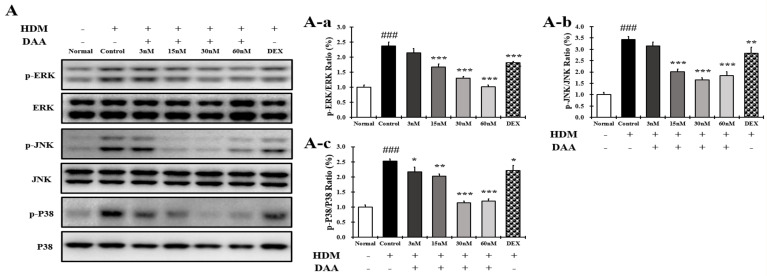
Effect of DAA on MAPK phosphorylation and NF-kB activation in EOL-1 cells. (**A**) DAA inhibited the phosphorylation of MAPK (ERK, JNK, and P38) in EOL-1 cells (**A-a**–**A-c**). (**B**) DAA also inhibited the degradation of IkBα and the translocation of NF-kB into the nucleus (**B-a**,**B-b**). IκBα was normalized to β-actin and NF-kB was normalized to Lamin B1. The results are expressed as the mean ± SD (n = 4). ### *p* < 0.001 vs. normal; * *p* < 0.05, ** *p* < 0.01, and *** *p* < 0.001 vs. control. DAA was used at 3 nM, 15 nM, 30 nM, and 60 nM. DEX, the positive control, was used at 2 nM.

**Table 1 molecules-26-03298-t001:** Content of DAA and AA in the *Noni* and *F.*
*noni* extract powder.

Content of DAA and AA in the Noni/*F. noni* Extract Powder (mg/g)
Sample	DAA	AA
*Noni*	10.37 ± 0.94	0.98 ± 0.08
*F. noni*	13.11 ± 0.25	1.40 ± 0.08

**Table 2 molecules-26-03298-t002:** Sequences of primers used in RT-qPCR.

Gene	Forward	Reverse
IL-1β (h)	AAA CAG ATG AAG GTG CTC CTT	TGG AGA ACA CCA CTT GTT GC
IL-6 (h)	AAA TTC GGT ACA TCC TCG ACG GCA	AGT GCC TCT TTG CTG CTT TCA CAC
IL-8 (h)	AAG CTG GCC GTG GCT CTC TTG	AGC CCT CTT CAA AAA CTT CTC
IL-25 (h)	CGA CCC AGA TTA GGT GAG GA	TCC ATC TTC ACT GGC CCT AC
IL-33 (h)	ACA GAA TAC TGA AAA ATG AAG CC	CTT CTC CAG TGG TAG CAT TTG
TNF-α (h)	AGG ACG AAC ATC CAA CCT TC	TTT GAG CCA GAA GAG GTT GA
TSLP (h)	TAG AGT GGG ACC AAA AGT ACC G	GGG ATT GAA GGT TAG GCT CTG G
MCP-1 (h)	GTC TCT GCC GCC CTT CTG T	TTG CAT CTG ATG GCA GTA GCT
TARC (h)	CTT CTC TGC AGC ACA TCC	AAG ACC TCT CAA GGC TTT G
MDC (h)	AGG ACA GAG CAT GGA TCG CCT ACA GA	AAT GGC AGG GAG GTA GGG CTC CTG A
RANTES (h)	CCG CGG CAG CCC TCG CTG TCA TCC	CAT CTC CAA AGA GTT GAT GTA CTC C
GAPDH (h)	TCG ACA GTC AGC CGC ATC TTC TTT	ACC AAA TCC GTT GAC TCC GAC CTT

## Data Availability

The data presented in this study are available on request from the corresponding author.

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
