# Peer review of "Effects of Deacetylasperulosidic Acid on Atopic Dermatitis through Modulating Immune Balance and Skin Barrier Function in HaCaT, HMC-1, and EOL-1 Cells"

_molecules, 2021, doi:10.3390/molecules26113298_

Round 1

Reviewer 1 Report

Dear authors,

Thank you for this well-rounded manuscript.

Overall, the manuscript is ready for publication, except that the spelling of "asperulosidic" and "deacetylasperulosidic" is inconsistent throughout the text. Sometimes, the name of these molecules is misspelled as "asFerulosidic" or "deacetylasFerulosidic". Please, review and make sure the correct spelling is used in the manuscript. The first example of misspelling is found in the title.

Best,

Reviewer.

Author Response

The final file will be uploaded after the answer is completed.
Since it is not possible to upload the cover letter at the same time, I added it to the front.

Modified both with deacetylasperulosidic acid and asperulosidic acid.

  1. Effects of Deacetylasperulosidic acid on atopic dermatitis through modulating immune balance and skin barrier function in HaCaT, HMC-1, and EOL-1 cells (Title)
  2. deacetylasperulosidic acid (Line 20 of text)
  3. Deacetylasperulosidic acid (Keyword)
  4. deacetylasperulosidic acid (Line 99 of text)
  5. asperulosidic acid (Line 108 of txet)
  6. deacetylasperulosidic acid and asperulosidic acid ( Line 575 of text)

The revised paper was revised by including the answer from Reviewer 2.

Reviewer 2 Report

Su Oh et al., studied the “Effects of Deacetylasferulosidic acid on atopic dermatitis through modulating immune balance and skin barrier function in HaCaT, HMC-1, and EOL-1 cells.” Atopic dermatitis (AD) is an inflammatory skin disease, which mediated via Th2 cytokine response. Considering the anti-AD potential of Noni (Morinda Citrifolia) in in vivo models, the authors have hypothesized that the active constituent of Noni, i.e., DAA and AA, might exhibit inducing anti-inflammatory response. To confirm the same, the authors have extracted, purified (by HPLC), and evaluated on in vitro models.

The study is very interesting and identified the anti-atopic dermatitis potential of DAA and AA on three cells lines. First of all, the selection of these three cell lines as a model of AD is very wise. The data appears promising. The material and methods provided are useful for repeatability.

Comments and suggestions

What are the IC50 values for these compounds?

 How much noni is required to reach the therapeutic concentration of DAA in human? Please shed light on this point.

Line 177-179: We can’t rule out the value of AA, although it is less. AA has equally shown an inhibitory effect on some cytokines, so it would be great if the authors evaluate some endpoint readout with AA in the remaining assays.

I am wondering, how the authors have omitted some important cytokines for evaluation, such as IL-22, IL-24. Please refer to these https://doi.org/10.37349/ei.2021.00005; https://doi.org/10.37349/ei.2021.00002

Figure 7: Why authors have not provided the IL-33 cytokine expression, although gene expression have provided.

Minor

Figure 9B: Please clearly explain in the results “cytosolic NF-kB and nucleus NF-kB".

Figure 10C: It looks strange, DEX group has not shown significance. Please re-check.

Line 43: Do authors mean “closely related to Th2 immune response” instead of AD. Please check.

Line 89-90: including neurological disorders. Please cite this https://doi.org/10.1016/B978-0-12-817780-8.00019-0

Please provide the full form of DEX (dexamethasone) at the first instance in the manuscript, although it provided in the material section.

Always use RT-qPCR, not qRT-PCR.

Gene names should be italics throughout the manuscript.

These two references are useful in the discussion.

https://www.nature.com/articles/cddis201790

https://www.sciencedirect.com/science/article/pii/S0091674920314081#undfig1

Line 441-442: Please rewrite the sentence.

Author Response

on this page, a cover letter has been added for the reply.

Round 2

Reviewer 2 Report

Su Oh et al., revised the manuscirpt by considering reviewers comments. 

The present manuscript is suitable for publication.